

# A glaciochemical study of 120 m ice core from Mill Island, East Antarctica

Mana Inoue[1,2], Mark A. J. Curran[1,3], Andrew D. Moy[1,3], Tas D. van Ommen[1,3], Alexander D. Fraser[1,4], Helen E. Phillips[2], and Ian D. Goodwin[5]

[1]Antarctic Climate & Ecosystems Cooperate Research Centre, Private Bag 80, University of Tasmania, Hobart, Tasmania 7005, Australia
[2]Institute for Marine & Antarctic Studies, University of Tasmania, Hobart, Tasmania 7001, Australia
[3]Australian Antarctic Division, Channel Highway, Kingston, Tasmania 7050, Australia
[4]Institute of Low Temperature Science, Hokkaido University, N19, W8, Kita-ku Sapporo 060-0819, Japan
[5]Marine Climate Risk Group, Department of Environment and Geography, Macquarie University, Eastern Road, New South Wales 2109, Australia

*Correspondence to:* Mana Inoue (manainoue@gmail.com)

**Abstract.** A 120 m ice core was drilled on Mill Island, East Antarctica (65° 30' S, 100° 40' E) during the 2009/2010 Australian Antarctic field season. Contiguous discrete 5 cm samples were measured for hydrogen peroxide, water stable isotopes and trace ion chemistry. The ice core was annually dated using a combination of chemical species and water stable isotopes. The Mill Island ice core preserves a climate record covering 97 years from 1913 to 2009 C.E., with a mean snow accumulation

of 1.35 metres (ice-equivalent) per year (mIE/yr). This northernmost East Antarctic coastal ice core site displays trace ion concentrations that are generally higher than other Antarctic ice core sites (e.g., mean sodium levels were 254 $\mu Eq/L$). The trace ion record at Mill Island is characterised by a unique and complex chemistry record with three distinct regimes identified. The trace ion record in Regime A displays clear seasonality from 2000 to 2009 C.E.; Regime B displays elevated concentrations with no seasonality from 1934 to 2000 C.E.; and Regime C displays relatively low concentrations with seasonality from 1913

to 1934 C.E. Sea salts were compared with instrumental data, including atmospheric models and satellite-derived sea ice concentration, to investigate influences on the Mill Island ice core record. The mean annual sea salt record does not correlate with wind speed. Instead, sea ice concentration to the east of Mill Island likely influences the annual mean sea salt record. A mechanism involving formation of frost flowers on sea ice is proposed to explain the extremely high sea salt concentration. The Mill Island ice core records are unexpectedly complex, with strong modulation of the trace chemistry on long timescales.

# 1  Introduction

The IPCC 5th Assessment Report states that there are insufficient Southern Hemisphere climate records to adequately assess climate change in much of this region. Ice cores provide excellent archives of past climate, as they contain a rich record of past environmental tracers archived in trapped air and precipitation. However Antarctic ice cores, especially those from East Antarctica, are limited in quantity and spatial coverage. To help address this, a 120 m ice core was drilled on Mill Island, East



Antarctica (65° 30' S, 100° 40' E). Hidrogen peroxide ($H_2O_2$), water stable isotopes ($\delta^{18}O$ and $\delta D$), and trace ion chemistry were measured from the 120 m Mill Island ice core. This study presents these measurement results.

Mill Island is a small island ($\sim 45 \times 35$ km), rising $\sim 500$ m above sea level, located in East Antarctica. It is connected to the Antarctic continent by the Shackleton Ice Shelf. The relatively low elevation and close distance to the ocean suggests the potential for significant input of maritime air to the snow falling at Mill Island. Mill Island is located approximately 500 km west of Law Dome, 350 km east of Mirny Station, 60 km north of the exposed rock formation known as Bunger Hills, and lies at the northern edge of the Shackleton Ice Shelf in Queen Mary Land (Fig. 1).

Mill Island is the most northerly Antarctic ice core site outside of the Antarctic Peninsula, and therefore the Mill Island ice core comprises the most northerly climate record for East Antarctica (Roberts et al., 2013). Mill Island experiences a polar maritime climate and high precipitation, particularly on its eastern flank, due to moist and warm air masses from the Southern Ocean brought onshore by low pressure systems. The site also experiences dry and cold air associated with strong katabatic winds from the continent and low level cloud, fog and rime formation over the summit caused by localised summer sea-breezes associated with nearby sea-ice breakout (Roberts et al., 2013). Records from Mirny Station show that the monthly mean temperature is below zero throughout the year (Turner and Pendlebury, 2004), suggesting that at its ~500 m summit elevation, Mill Island likely experiences little melt.The high precipitation rate and minimal melt makes Mill Island an ideal site from which to extract high resolution climate records for the Southern Hemisphere.

In the 2008/2009 austral summer, one shallow core (MIp0809) was recovered during a reconnaissance expedition. The main ice core drilling campaign was carried out during the 2009/2010 Australian Antarctic program. The team spent three weeks in the field, and drilled one 120 m main ice core (MI0910) and seven shallow (from ~5 m to 10 m) firn cores. This paper focuses on the main (MI0910) 120 m ice core record that is supplemented by two shallow firn cores MIp0910 and MIp0809 (Table 1). The 120 m ice core was drilled using the intermediate-depth ice core drill (ECLIPSE ice coring drill, Icefield Instruments, Inc.). A $\sim 2$ m trench was excavated prior to drilling. Thus, the top 2 m of the full record presented here is obtained from the MIp0910 core.

Early studies attributed the main source of sea salts in ice cores to sea spray from the open ocean, transported by strong winds associated with storm events (e.g., Wagenbach, 1996; Legrand and Mayewski, 1997; Curran et al., 1998; Wagenbach et al., 1998a). More recently, Rankin et al. (2002) and Kaleschke et al. (2004) reported the importance of frost flowers (sea salt crystals which form on new sea-ice) as a sea salt source. Frost flowers have a sea salt concentration three times higher than sea water (Perovich and Richter-Menge, 1994; Wolff et al., 2003; Kaleschke et al., 2004). Hence frost flower-originated aerosols contain a higher concentration of sea salt than aerosols originating from sea water. Yang et al. (2008) also suggested the sublimation of salty blowing snow on sea ice as a potential unfractionated sea salt source. It is likely that different sea salt sources dominate and contribute to the sea salt records at different sites (Abram et al., 2011).

The aims of this paper are to present a high resolution, well-dated records of water stable isotopes ($\delta^{18}O$, $\delta D$), and trace ion chemistry (sea salts, sulphate, methanesulphonic acid) at Mill Island, and to investigate the seasonal and interannual variability of sea salt, in order to reveal the climate factors that influence the Mill Island ice core record. This was completed by investigating the characteristics of the trace chemistry record, and by examining the environmental factors that influence these records,



e.g., wind speed and direction, sea ice configuration, and deposition processes. The sodium ($Na^+$) and sulphate ($SO_4^{2-}$) records were determined to represent sea salt contribution to the Mill Island site.

## 2 Method

### 2.1 Ice core processing

The Mill Island firn and ice cores were processed in a clean freezer laboratory using similar techniques to those described by Curran and Palmer (2001). The density of the cores were computed using core diameter, length and weight measurements. Visual observation was also completed for stratigraphy studies. The cores were then transversely divided into three sticks using a clean band-saw. The sticks were used for hydrogen peroxide, stable water isotopes, and trace ion chemistry measurements. The sticks for hydrogen peroxide and water stable isotope measurements were then cut into 4 cm length samples. The central

sticks for trace ion chemistry were cleaned to avoid contamination and sampled every 4 cm (i.e., approx. 25 samples per ∼ one m core segment). Cleaning was achieved by removing ∼ three mm of each surface with a microtome under a laminar airflow hood. Chemistry samples were stored in a Coulter cup (Kartell brand), melted in a refrigerator, and then refrozen again to minimise methanesulphonic acid (MSA) loss (Abram et al., 2008). The refrozen samples were melted prior to analysis. All tools used for processing ice cores were carefully pre-cleaned with deionised ultra-clean Milli-Q water (resistivity > 18

MΩ–cm), and polyethylene gloves were worn during the ice core processing to minimise contamination.

### 2.2 Sample measurement

Hydrogen peroxide ($H_2O_2$) measurements were carried out using a fluorescence detector as detailed by van Ommen and Morgan (1996). Four cm length samples were analysed at 8 cm resolution from the surface to a depth of 25 m, and then at a sample resolution of 12 cm for the remainder of the 120 m ice core.

Water stable isotopes ($\delta^{18}O$ and $\delta D$) were measured using a Eurovector EuroPyrOH HT elemental analyser interfaced in continuous flow mode to an Isoprime isotope ratio mass spectrometer. Samples at four cm resolution were melted in a refrigerated unit prior to analysis. Liquid samples were sampled by a Eurovector liquid auto-sampler (LAS EuroAS300). Analytical precision for $\delta D$ is < 0.5 ‰ and for $\delta^{18}O$ is < 0.1 ‰, and values are expressed relative to the Vienna Standard Mean Ocean Water 2 (VSMOW2). Deuterium excess (D-ex) was then calculated from the measured $\delta D$ and $\delta^{18}O$ using the following

equation

$$\text{D-ex} = \delta D - 8 \times \delta^{18}O \quad \text{(Paterson, 1994)} \tag{1}$$

Trace ion chemical measurements were carried out using a suppressed ion chromatograph (IC) as detailed by Curran and Palmer (2001). Samples were melted overnight in a refrigerator prior to analysis. Due to the high sea salt concentration, the melted samples were diluted at a ratio of 50:1 in autosampler polyvials using a micropipette within a laminar flow hood. Further

dilutions (5 to 100 times) were completed according to the sea salt concentrations depending on the initial results. Samples were then analysed using a Dionex® AS18 ICS-3000 (2 mm) microbore ion chromatograph. The major ion species measured



in this study were mehanesulphonic acid ($CH_3SO_3^-$ [MSA]), chloride ($Cl^-$), nitrate ($NO_3^-$), sulphate ($SO_4^{2-}$), sodium ($Na^+$), potassium ($K^+$), magnesium ($Mg^{2+}$), and calcium ($Ca^{2+}$). Anions (i.e., MSA, $Cl^-$, $SO_4^{2-}$, and $NO_3^-$) were analysed using an IonPac® AS18 separation column and AG18 guard column. Cation (i.e., $Na^+$, $K^+$, $Mg^{2+}$, and $Ca^{2+}$) analysis was performed using CS12A separation columns. The system performed anion and cation analysis simultaneously using dual isocratic pumps. The non sea salt sulphate ($nssSO_4^{2-}$) record was then calculated using the formula

$$[nssSO_4^{2-}] = [SO_4^{2-}] - k_{Na} \times [Na^+] \tag{2}$$

where $k_{Na}$ is sea salt ratio of $SO_4^{2-}$ to $Na^+$, 0.120 (Mulvaney and Wolff, 1994). All trace ions were calibrated using diluted standards (Curran and Palmer, 2001) expressed in concentrations of micro equivalents per litre ($\mu Eq/L$).

## 2.3 Datasets

### 2.3.1 Wind direction and wind speed

Due to a lack of in-situ meteorological observation data available at Mill Island, atmospheric model outputs were used to investigate the differences between regimes. Wind data were derived from National Centers for Environmental Prediction (NCEP) Climate Forecast System Reanalysis (CFSR) (Environmental Modeling Center, 2010). CFSR provides high resolution atmospheric reanalysis data ($\sim 0.313$ degree $\times$ 0.312 degree). The closest grid point to Mill Island was chosen for this analysis (65° 24' 42.84" S, 100° 56' 15" E, $\sim 17$ km east of the exact MI0190 drilling site). CFSR data are available from 1979.

### 2.3.2 Sea ice concentration

Sea ice concentration data were provided by the National Snow & Ice Data Center. Sea ice concentration was derived from the passive microwave Scanning Multichannel Microwave Radiometer (SMMR) instrument on the Nimbus-7 satellite, and from the Special Sensor Microwave/Imager (SSM/I) instruments on the Defense Meteorological Satellite Pro- gram's (DMSP) -F8, -F11, and -F13 satellites, using the bootstrap algorithm (Comiso, 2000, updated 2014). The data are provided at a monthly time step and have a spatial resolution of 25 km. Sea ice concentration data are available from 1979.

## 3 Ice core dating

MI0910 and MIp0910 were dated by counting annual layers using $H_2O_2$, water isotopes ($\delta^{18}O$, $\delta D$), and deuterium excess (D-ex) according to the methods presented in Plummer et al. (2012). The layer counting method using this multi-proxy approach was subsequently confirmed by the non sea salt sulphate ($nssSO_4^{2-}$) record, that matches the timing of volcanic eruptions at Law Dome (LD) (Plummer et al., 2012) and at other ice core sites (Cole-Dai et al., 1997, 2000). The shallow cores MIp0910 and MIp0809 were also annually dated using the layer counting technique to supplement the top of the MI0910 core (Table 1), and to verify the MI0910 dating.

MIp0910 covers four years, from 2006 to 2009 C.E. (Fig. 2) and there is good agreement with the $\delta^{18}O$, D-ex, $Na^+$, and $SO_4^{2-}$ where MIp0910 overlaps with the top of the MI0910. MIp0809 was dated by counting annual layers of $\delta^{18}O$ and D-ex





($H_2O_2$ measurements were not available for MIp0809). MIp0809 covers 15 years, from 1994 to 2008 C.E. (Fig. 2). The overlap of $\delta^{18}O$ and D-ex for MIp0809 is in agreement with MI0910. These comparable and overlapping records provide confidence in a continuous ice record from surface to a depth of 120 m. Moreover, the fact that these two individual nearby ice core records (10 km between MIp0809 and MI0910 sites) acquired in different years and processed independently show similar records

provides confidence in the ice core dating methodology used.

      The ambiguities in the seasonal cycles of $H_2O_2$ and $\delta^{18}O$ give rise to potential dating errors. Such errors are statistically independent, as the decision of counting a seasonal cycle as a year marker is not affected by other errors. Thus instead of adding each error linearly, the errors can be combined in quadrature (e.g., error = $\sqrt{error1^2 + error2^2 + error3^2...}$ [Barlow (1989)]). Because the dating is confirmed by the volcanic record, the errors are periodically set to zero at the timing of each

major eruption year (e.g., 1991, 1984, and 1963). As a result, MI0910 dating error is within the range from +2.4 to - 3.5 years.

## 4   Results

### 4.1   Peroxide and water stable isotopes

The $H_2O_2$ record generally shows a strong annual cycle, except for the late 1970s and early 1950s where there is an observed loss of $H_2O_2$ (Fig. 3 a). $H_2O_2$ seasonality loss is attributed to transient melt events (van Ommen and Morgan, 1996). However

this is not necessarily the case at Mill Island. Further discussion of this loss of $H_2O_2$ is presented later. The $H_2O_2$ record shows a baseline drift prior to 1935 C.E. which is attributed to calibration problems. Despite this, the data show strong seasonal variations which are sufficient to assist annual layer counting throughout most of the record. The water stable isotope ($\delta^{18}O$, $\delta D$) and the deuterium excess (D-ex) records also show annual cycles throughout the core (Figs. 3 b, c, and d)

      A water vapour diffusion correction was computed using the method adopted in van Ommen and Morgan (1997) and Sinclair

et al. (2012), with specific Mill Island parameters (density profile, mean temperature, and atmospheric pressure). As a result, the diffusion length reached 6.7 cm at a depth of 43 m in the firn. With the high snow accumulation rate in the Mill Island ice core (1,312 kg/m²yr, Roberts et al., 2013), this is small enough to ignore.

      Figure 4 shows the average seasonal cycles of (a) $H_2O_2$, (b) $\delta^{18}O$, (c) $\delta D$, and (d) D-ex for the 97 yr MI0910 ice core record. The monthly mean of each species was computed by linearly dividing each year data point into 12 portions. The average

seasonal cycles are presented over two years to demonstrate the structure of both the summer and winter features.

      The $H_2O_2$ average seasonal cycle shows a peak in summer with a maximum value in December, and a trough in winter with a minimum value in June (Fig 4 a). This timing coincides with the timing of the solstices, due to the requirement of sunlight for $H_2O_2$ production.

      The average seasonal cycles of $\delta^{18}O$ and $\delta D$ show a peak in summer and a trough in winter, with a maximum in January

and a minimum in August (Fig 4 b and c, respectively). The $\delta^{18}O$ seasonal cycle is compared with the temperature record from Mirny Station. The Mirny Station seasonal temperature record shows a maximum in January and a minimum in August. However the "coreless" winter pattern observed at Mirny Station, which is a broad temperature minimum during winter without a defined trough, was not observed in the Mill Island isotope records (Fig. 4 b, and c).



The D-ex seasonal cycle shows a minimum in summer (December), and a maximum in winter (May) (Fig 4 d). There is a four month lag in D-ex compared to $\delta^{18}O$. This lag is also reported for the LD ice core (Delmotte et al., 2000). The D-ex record is used as a proxy for the moisture source temperature and sea ice cover (Delmotte et al., 2000; Uemura et al., 2012). However D-ex is affected by the relative humidity of the oceanic source region as well as by sea surface and polar temperatures

(Landais et al., 2008; Uemura et al., 2008). Measurement of $\delta^{17}O$ is required for proper interpretation of D-ex (Landais et al., 2008; Winkler et al., 2012). However $\delta^{17}O$ was not measured for the Mill Island ice core due to laboratory limitations.

## 4.2 Trace chemistry record

Figure 5 shows concentrations of a) $Na^+$, b) $Cl^-$, c) MSA, d) $SO_4^{2-}$, and Fig. 6 shows concentrations of a) $nssSO_4^{2-}$, b) $Mg^{2+}$, c) $Ca^{2+}$, and d) $NO_3^-$ for the entire ice core. Typically, these trace ion species show strong seasonal variations (e.g., $Na^+$ and

10 $Cl^-$ have a winter peak, MSA has a summer peak). However the results for the trace ion chemistry at Mill Island show clear seasonality only in the top 10 years of the ice core. The seasonality in trace chemistry either disappears, or shows incoherent peaks prior 2000 C.E. (Fig. 5). The baselines of $Na^+$ and $Cl^-$ are also higher from 1934 to 2000 C.E.. The $Na^+$ and $Cl^-$ chemistry seasonality from 2009 to 2001 disappears between $\sim 2000$ and $\sim 1934$, to be replaced by high baseline values. Prior to 1934 the seasonality is present again and values are lower in concentration than in the other period. These periods

(2009 – 2001, 2000 – 1934, 1933 – 1913 C.E.) are shown in Fig. 6 and henceforth termed regimes A, B and C, respectively. Further discussion about these regime changes is presented later.

Average seasonal cycles (1913 – 2009 C.E.) of trace ion species are displayed in Fig. 7. Despite the unclear seasonality prior to 2001 C.E. (Fig. 6), the average seasonal cycles of $Na^+$ and $Cl^-$ show clear seasonal variability with a peak in winter (May), and a trough in summer (December) (Fig. 7a and b, respectively).

The MSA average seasonal cycle shows low concentration during winter (May - October), then peaks in spring (November) and autumn (April) (Fig. 7c). However during summer (December - March), concentrations are relatively low. There is a reversed phase observed between $Cl^-$ and MSA during the latest 10 years, however they are synchronised in older parts of the record (e.g., between $\sim 1965$ to $\sim 1975$ C.E.). This is likely due to post-deposition MSA movement (Curran et al., 2002).

Sulphate ($SO_4^{2-}$) also shows clear seasonal variability with a peak in April, and a trough in November (Fig. 7, d). The winter

time maximum in the $SO_4^{2-}$ record indicates that sea salt is the dominant source of the $SO_4^{2-}$ at this site.

The mean concentration of $nssSO_4^{2-}$ is negative and the average seasonal cycle also shows a negative value throughout the year (Fig. 7e). This indicates that the $nssSO_4^{2-}$ at Mill Island is highly fractionated by sea-salt. Thus, a different $k$ value is needed to correctly derive $nssSO_4^{2-}$. Investigation of the specific $k$ value and sea salt fractionation are discussed later.



## 5 Discussion

### 5.1 Sea salt regimes at Mill Island

Time series of $Na^+$, $SO_4^{2-}$, $\delta^{18}O$, and D-ex are shown over the period from 1913 to 2009 C.E. (Fig. 8). $Na^+$ shows clear differences between the regimes. The $Na^+$ winter (April to October) peak during regime A is not present in regime B. Instead, $Na^+$ in regime B shows lengthy "plateau" periods ($\sim 300$ $\mu Eq/L$), and "valley" periods, which have a relatively low concentration ($< 100$ $\mu Eq/L$). $Na^+$ in regime C shows lower concentrations ($\sim 30$ $\mu Eq/L$) with observable seasonality (except for 1917 – 1920 C.E.).

The $SO_4^{2-}$ record shows peaks in winter during regime A, and seasonally-incoherent high concentration peaks ($> \sim 40 \mu Eq/L$) in regime B. Regime C, however, differs to regimes A and B, with low concentrations of $SO_4^{2-}$ ($\sim 5$ $\mu Eq/L$). In regime B, $SO_4^{2-}$ shows occasional winter peaks, e.g., between 1934 and 1940 C.E.; 1950 and 1957 C.E.; and 1977 and 1987 C.E.. The winter peaks in $SO_4^{2-}$ in regime B suggest that the main source of $SO_4^{2-}$ is sea salt. When the $SO_4^{2-}$ record shows a high winter concentration ($> \sim 40$ $\mu Eq/L$), the $Na^+$ record plateaus.

For the $\delta^{18}O$ ratio, other than higher values during summer (December – January) after 2000 C.E., there appears to be little difference between regimes. D-ex also does not show any differences associated with the regimes. Although, lower values are observed during winter before 1950 C.E. than after 1950 C.E.. It appears that the regime shifts are only evident in the sea salt trace ion record. Chloride ($Cl^-$, not shown) also shows features similar to the $Na^+$ record, i.e., clear seasonality in regime A, mix of "plateau" and "valley" regions in regime B, and lower concentrations with observable seasonality in regime C.

Average seasonal cycles of $Na^+$, $SO_4^{2-}$, $\delta^{18}O$, and D-ex for each regime are shown in Fig. 9. $Na^+$ and $SO_4^{2-}$ both show seasonality with a winter peak in regimes A (blue line) and B (green line). The variability of the $Na^+$ concentration is lower in regime B (minimum 214 $\mu Eq/L$ in November, maximum 293 $\mu Eq/L$ in April) compared with regime A (minimum 92 $\mu Eq/L$ in December, maximum 1,222 $\mu Eq/L$ in May), and the seasonality is not as clear (Fig. 9 a). The $Na^+$ variation and concentration was lowest during regime C (minimum 17 $\mu Eq/L$ in January, maximum 55 $\mu Eq/L$ in July), and seasonality is still evident, with a peak in winter. However $SO_4^{2-}$ seasonality is not evident in regime C.

$\delta^{18}O$ (Fig. 9 c) shows enriched values ($\sim -11$ ‰) during summer in regime A compared to regimes B and C (Fig. 9 c). This suggests that the controlling influence on $\delta^{18}O$ (i.e., temperature) at the coring site has increased during summer since $\sim 2000$ C.E.. The D-ex seasonal cycle shows the most depleted values ($\sim 5$ ‰) during summer in regime A. In regime C, the D-ex variability within a year is smaller than in other regimes (Fig. 9 d). This may indicate that the moisture source has changed, or some changes have happened in the moisture source region since regime C (or since 1950 C.E., according to Fig. 8).

In summary,

Regime A: Clear seasonality with a winter peak is observed for $Na^+$ and $SO_4^{2-}$. The mean concentrations of $Na^+$ and $SO_4^{2-}$ are high (451 $\mu Eq/L$ and 30.2 $\mu Eq/L$, respectively) (Fig. 8, 9).

Regime B: $Na^+$ shows "plateaus" of $\sim 300$ $\mu Eq/L$ and "valleys" (shorter periods of lower concentration, $< 100$ $\mu Eq/L$). $SO_4^{2-}$ shows seasonally-unaligned peaks ($> \sim 40$ $\mu Eq/L$) during which $Na^+$ "plateaus" (Fig. 8).



Regime C: Mean concentrations of $Na^+$ and $SO_4^{2-}$ are low (32.5 *μEq/L* and 5.6 *μEq/L*, respectively). $Na^+$ shows winter time peak, but no seasonality is observed in the $SO_4^{2-}$ record (Fig. 8, 9).

## 5.2 Sea salt regime changes and the stratigraphy of MI0910

The regime changes only influence the trace ion record. $\delta^{18}O$ variability shows no detectable changes during the observed regime changes other than enriched values in summer after 2001 C.E., and D-ex shows lower values prior to 1950 C.E.. The possible reasons for the trace ion record features include analytical error in measurement or methodology; snow/firn melt; or a true environmental signal:

### 5.2.1 Possibility of analytical error in measurement or methodology

Repeat trace ion chemistry analysis was completed using different dilutions and this yielded the same results, thus discounting the possibility of errors due to analytical measurement. Additionally, the two shallow cores were analysed independently using the same instrument and method. These trace ion measurements from both shallow ice cores agree with the MI0910 record (Fig 2). Thus analytical error in measurement or methodology is not the cause of these features.

### 5.2.2 Possibility of snow/firn melt

The stratigraphy of the MI0910 ice core shows higher density layers distributed occasionally throughout the entire ice core (Roberts et al., 2013). These layers may be due to melt, however the cause of each layer is difficult to explicitly investigate by close inspection alone (Kinnard et al., 2008). Thus all such layers observed in MI0910 are here termed "crust layers" for convenience. Visual stratigraphy observation was achieved by counting and logging the crust layers.

Figure 10 shows the distribution of crust layers observed in the ice core (blue vertical lines) along with the full records of $H_2O_2$, $Na^+$, and $SO_4^{2-}$. A total of 172 crust layers were recorded. The crust layers appear not to correspond with the periods of observed loss of $H_2O_2$ or reduced seasonality in the trace ion record. For example, the early 2000s (indicated with a grey ellipse labelled "a") includes multiple crust layers, but all three species ($H_2O_2$, $Na^+$, and $SO_4^{2-}$) show clear seasonality. Between the late 1970s and early 1980s (grey ellipse "b") there is a loss of $H_2O_2$ seasonality, however, this period includes occurrences of both few crust layers (late 1970s) and many crust layers (early 1980s). Another period of observed loss of $H_2O_2$ seasonality occurs during the early 1950s (grey ellipse "c") yet this period shows few crust layers. The trace ion record, grey ellipses "b" and "c" show similar characteristics for each species, whereby there is a high concentration of $Na^+$ with a muted seasonal signal and a high concentration of $SO_4^{2-}$. However there is no relationship with crust occurrence frequency and the defined regimes.

The observed $H_2O_2$ loss may be related to $SO_4^{2-}$ concentration. $H_2O_2$ is believed to be the most efficient oxidizing agent of $SO_2$, producing $SO_4^{2-}$ (Laj et al., 1990). In the grey ellipses "b" and "c" (Fig. 10), large peaks of $SO_4^{2-}$ are associated with the depletion of $H_2O_2$. However, not all $SO_4^{2-}$ peaks are associated with $H_2O_2$ loss. Nitrogen oxides also tend to reduce the



concentration of peroxide (Sigg and Neftel, 1991). However, there are no associated nitrate features observed in the record (see Fig. 6). The reason for the absence of the $H_2O_2$ peaks in these two regions is unknown.

In the polar snowpack percolation zone, melt events occur generally during summer (Langway, 1970). Assuming that all summer crust layers are caused by melt events, there is an implication that these events are associated with temperature (hence

$\delta^{18}O$). The crust layers during the summer period (October – March) and summer mean $\delta^{18}O$ were compared.Thirty-four summers out of the 97 year record had crust layers, and the data points were not normally distributed (not shown), thus a Spearman's rank correlation was used to asses the correlation. There was no significant correlation between the number of melt layers and the associated summer mean $\delta^{18}O$ ($\rho = -0.08$, p = 0.62, n = 34). This indicates that these crust layers may not be melt layers. Furthermore, the stratigraphy showed no sign of strong melt.

Strong wind may cause of the crust layers (Alley et al., 1997). Wind and temperature data were derived from National Centers for Environmental Prediction (NCEP) Climate Forecast System Reanalysis (CFSR) (Environmental Modeling Center, 2010) due to no long term observation records being available on Mill Island. CFSR provides high resolution data ($\sim 0.313$ degree $\times 0.312$ degree, 6 hourly). The closest grid point to Mill Island is at 65° 24' 42.84" S, 100° 56' 15" E, which is 17 km east of the exact drilling point of MI0910. This point was chosen for this analysis. The data period for analysis is between 1979

and 2009 C.E.. Figure 11 shows monthly bins of crust layers, mean temperature, mean number of six-hourly wind speed data which exceed 15 m/s (i.e., high wind periods), and mean number of six-hourly wind speed data which are less than 5 m/s (i.e., low wind periods). Generally, there is higher wind speed during winter. Both the number of crust layers and the number of high wind periods peaks in June and July. This indicates that the crust layers identified during winter may have been formed by strong wind events (Alley et al., 1997).

Additionally, fog events could be a cause of the crust layers. Fog and rime accretion associated with the fog events were observed during the field season at Mill Island (M. Curran, personal communication 2014). This rime deposition may appear as low-density crust layers (Alley et al., 1997). However, fog and low cloud events are difficult to accurately retrieve from atmospheric model output data (Inoue et al., 2015). An AWS instrumented with shortwave and longwave radiometers (in addition to standard components, relative humidity sensor, wind speed/direction/mean sea level pressure/precipitation) would provide

an ideal tool to assess the occurrence of fog events at the Mill Island site. Fine 1 cm sample resolution isotope measurements from the Mill Island shallow core show no apparent influence of crust layers on the record (not shown). Thus the crust layers probably have a minimal impact on the chemical interpretation of the Mill Island ice core records.

### 5.2.3 The possibility of true environmental signals

Since both analytical errors in measurement or methodology and snow/firn melt were discounted as the cause of the ambiguous

sea salt seasonality, the three different regimes identified may indicate the recording of different environmental signals at Mill Island. The influence of true environmental signals on the chemistry record at Mill Island is explored in the next section.



### 5.3 Influence of environmental signals on sea salt record

Sea ice and atmospheric reanalysis data were compared with the Mill Island sea salt record to investigate the possibility of true environmental signals in the regime change. Both sea ice concentration and atmospheric reanalysis data are available only since ∼ 1979. Thus only regimes A and B (after 1979) are investigated in the next section.

### 5.3.1 Wind direction and wind speed at Mill Island

Many ice core studies suggest that sea salt is a proxy for wind and storminess (e.g., Wagenbach, 1996; Legrand and Mayewski, 1997; Curran et al., 1998), because salt is transported by air mass movement. Thus, wind direction and speed are investigated in this section to determine the Mill Island sea salt transport mechanism.

The Mill Island wind rose climatology, created from the 6-hourly wind speed and direction data, was generated using data from 1979 to 2009 C.E.. At Mill Island, the wind direction is predominantly from the east, and the mean wind speed over the period is 7.6 m/s (see Fig. 12).

The relationship between wind speed and $Na^+$ and $SO_4^{2-}$ concentration was investigated by correlating annual, summer (October – March) and winter (April – September) means of $Na^+$ and $SO_4^{2-}$ concentration against the number of data points where the wind speed was < 5 m/s, 5 – 15 m/s, and > 15 m/s in the associated period (see Table 2). There is no significant correlation between annual mean $Na^+$ concentration and annual mean wind speed. The number of data points per year where the wind exceeds 15 m/s, or less than 5 m/s also show no significant correlations with annual mean $Na^+$ concentration. There is a significant negative correlation between $Na^+$ concentration and number of data between wind speed 5 m/s – 15 m/s per year (r = − 0.51, p < 0.01). However this correlation is strongly influenced by two data point outliers in 2006 and 2007 C.E. where the $Na^+$ concentration is extremely high. The regression slope of the $Na^+$ concentration versus medium wind speed is low, indicating that this correlation displays little predictive power. This negative correlation between $Na^+$ and wind speed 5 m/s – 15 m/s is likely coincidental and thus disregarded. To confirm that this relation is coincidental, the number of data points per year with wind speed less than or more than 7 m/s also shows no correlation with $Na^+$ concentration. Similarly, the significant correlation between the data points and wind speed 5 m/s, 5 – 15 m/s, and > 15 m/s during winter is thought to be due to outlying data points in the winters of 2002, 2006, and 2007 C.E.. Thus, the wind speed is unlikely related to the Mill Island sea salt regime changes, at least post-1979 C.E.. Correlations between $SO_4^{2-}$ concentration and wind speeds show almost the same results, except the outliers occur in years 2002 and 2006 C.E..

Sixty percent of the wind at Mill Island comes from the easterly quadrant (wind direction between 45 and 135 degrees). Ninety-nine % of wind with speed greater than 15 m/s wind blows from east. This indicates that the sea salt source at Mill Island is predominantly from the east. Therefore, the next section focuses on the environment to the east of Mill Island.



### 5.3.2 Relationship between sea ice concentration and sea salt

Bowman Island (65° 12' S, 103° 00' E) is located ∼100 km east of Mill Island (Fig. 13). During the observation record, the ocean between Mill Island and Bowman Island is typically free of ice during summer, and covered with sea ice during winter. The sea ice cover in this area could possibly influence the sea salt record.

Monthly sea ice concentration (SIC) was investigated for the region between Mill Island and Bowman Island for the period between January 1979 and December 2009. At the 25 km resolution of the SIC dataset, there are five SIC pixels between Mill Island and Bowman Island (see Fig. 13 for coordinates and pixel name). Annual mean SIC in these pixels was compared with annual mean concentrations of $Na^+$ and $SO_4^{2-}$.

Table 3 shows the correlation coefficient between annual mean SIC and annual mean concentrations of $Na^+$ and $SO_4^{2-}$.
Annual mean SIC is negatively correlated with mean annual $Na^+$ concentration for all pixels except SIC-W. The highest correlation with $Na^+$ is at SIC-SE ($r = -0.57$, $p < 0.01$). The annual mean concentration of $SO_4^{2-}$ is also significantly anti-correlated with annual mean sSIC for all five SIC pixels. SIC-S shows the highest negative correlation ($r = -0.58$, $p < 0.01$). Thus, SIC values from SIC-S and SIC-SE were averaged to form a single record, termed SIC-m.

Table 4 shows the correlation coefficient between SIC-m and annual mean concentrations of $Na^+$ and $SO_4^{2-}$ between 1979
and 2009. SIC is significantly anti-correlated with both $Na^+$, and $SO_4^{2-}$ in all three periods. This indicates that the time series of annual mean sea salt record from Mill Island may represent sea ice concentration variability at the local area (Fig. 14).

Figure 15 shows time series of SIC-m, SIC-W, $Na^+$, and $SO_4^{2-}$ for the period between 1979 and 2009 C.E., covering all of regime A and approximately one third of regime B. The horizontal dashed blue line indicates the mean value of SIC-W, and dotted blue lines indicate the $1\sigma$ standard deviation of SIC-W. This figure clearly shows the negative correlation between
SIC-m and $Na^+$, $SO_4^{2-}$.

It has been shown that the Mill Island site shows high sea salt concentrations, the prevailing wind is from the east, and the sea ice to the east frequently shows low concentration. A mechanism to explain the relationship between wind direction, sea ice concentration, and high sea salt concentration is proposed here:

   1. Areas of open water between Mill and Bowman Islands (i.e., the SIC-m area) freeze to form new sea ice.

2.  (a) Frost flowers are produced on newly formed sea ice, then fragments are transported to Mill Island by the prevailing easterly wind (Hall and Wolff, 1998); or

        (b) sea salt-enriched brine migrates upward through sea ice brine channels to the snow on sea ice. Then the salty snow is blown to Mill Island by the prevailing easterly wind.

However, SIC-m is not particularly low in 2002, 2006, and 2007 C.E. when $Na^+$ and $SO_4^{2-}$ concentrations are high. The
lowest SIC-m years are in 1993 and 1994 (45.1 % and 46.4 %, respectively). In these years, $Na^+$ and $SO_4^{2-}$ show a mid-range concentration (weak anti-correlation between SIC and $Na^+$, $SO_4^{2-}$).

Differences between 1993 – 1994 and 2006 – 2007 C.E. are found in SIC-W. In 2006 – 2007 C.E., SIC-W concentration is within $1\sigma$ of the mean sea ice concentration (73.9 % in 2006 C.E. and 69.6 % in 2007 C.E.), whereas in 1993 – 1994 C.E.,





SIC is more than $1\sigma$ below the mean (58.0 % and 62.2 % respectively). This implies that the SIC-W may affect the sea salt transport process. Though in 2002 C.E., high levels of Na$^+$ are not clearly explained with this hypothesis alone.

Figure 16 shows a schematic diagram of a hypothetical sea salt transport mechanism. The edge of large ice-covered islands such as Mill Island typically exhibit a vertical discontinuity on the order of > 10 m, which may block the transport of direct sea spray and sea water aerosol particles onto the island (Fig. 16 a). If stable land-fast sea ice cover exists, it facilitates formation of a snow ramp, which effectively bridges the vertical gap between the land-fast sea ice and the ice sheet (Fig. 16 b). For example, when SIC-m is low and SIC-W is high (2006 C.E.), Mill Island records extremely high sea salt concentrations, because of an abundance of available sea salt as frost flowers in SIC-m area and as salty brine snow in SIC-m and SIC-W areas, as well as an effective mechanism to transport sea salt to Mill Island. When both SIC-m and SIC-W are relatively low (e.g., 2002 C.E.), Mill Island records still high Na$^+$ but not as high as the first case, because although there is an abundance of sea salt, the pathway (i.e., a snow ramp) is not present. When both SIC-m and SIC-W are high (2001, 1996, 1987 C.E.), Mill Island records low sea salt concentrations, suggesting that frost flowers are a more important sea salt source than the briny snow. This hypothesis is strengthened by noting that the ratio of SO$_4^{2-}$ to Na$^+$ in 2001 C.E. (0.121) and 1996 C.E. (0.122) is close to the sea water ratio of 0.12. When both sea ice concentration at SIC-m and SIC-W, and sea salt are high (e.g., 1991), the source of sea salt could be the briny snow on sea ice and/or nearby open water with storm events (the ratio of SO$_4^{2-}$ to Na$^+$ in 1991 C.E. is 0.114). This snow ramp theory works well for the vast majority of years.

Figure 17 shows an aerial photograph of an ice-capped island edge, adjacent to land-fast sea ice, covered with a well-formed snow ramp. The photo was taken on the $11^{th}$ of February, 1947 over Bowman Island. The circle (a) shows an example of the vertical discontinuity from sea level to the ice cap. The circle (b) demonstrates a well-formed snow ramp. The same feature is expected to form at Mill Island. Fraser et al. (2012) demonstrated the presence (and, at times, absence) of multi-year landfast ice to the east of Mill Island, which would facilitate snow ramp formation.

Figure 18 shows annual variations of SIC-m, SIC-W, Na$^+$ and SO$_4^{2-}$ concentrations for the period between 1979 and 2009 C.E.. Both SIC-m and SIC-W are generally high in early summer (December and January), and low in late summer (February and March). The negative correlation between SIC-m and trace ions can be seen on a monthly basis here. For example, SIC-m in early 1995 C.E. shows low concentrations (< ∼ 60 %) then becomes higher (> ∼ 70 %) later in the year. Na$^+$ shows high concentration (> ∼ 300 $\mu Eq/L$) in early 1995, then low concentration (< ∼ 100 $\mu Eq/L$) later in the year. Similar features are also seen in 2000. A case showing high SIC-m and SIC-W but low sea salt is observed from 1985 – 1987 C.E.. Some years show high sea ice coverage in SIC-W throughout the period, which suggests the existence of multi-year land-fast ice (e.g., 1986 – 1991, 1995 – 1997, and 2003 – 2006 C.E.).

### 5.3.3 Local ice shelf variability

The local sea ice concentration changes mentioned in Section 5.3.2 may be related to corresponding changes in the local ice shelf configuration (Massom et al., 2010). Using NASA Moderate Resolution Imaging Spectroradiometer (MODIS) satellite imagery, two major configuration changes were observed between 2000 and 2009 (Figs. 19 and 20).





The first event was the calving of the Scott Glacier in 2002 between Chungunov Island and Mill Island (Fig. 19 b), red dashed line). The event formed an iceberg named C20 (not shown), which drifted westward and continued to break up (Evers et al., 2013). This change occurred entirely to the west of Mill Island (i.e., downstream both oceanographically and atmospherically), thus is unlikely to have influenced the Mill Island record.

The next event was the export of Pobeda Ice Island (C5) from the north-north-west of Mill Island in 2003 or 2004 (Fig. 20). In the image from the $6^{th}$ of Mar, 2003 (Fig 20 a), a large tabular iceberg, Pobeda Ice Island, is grounded to the north-west of Mill Island. This iceberg is not present in the $15^{th}$ of Sep, 2004 image (Fig 20 b). The presence of such an ice island presents a strong dynamical barrier to mobile pack ice being advected westward in the coastal current, leading to higher pack ice concentration to the east of this barrier (Fraser et al., 2012).

These changes all occurred to the north or west of Mill Island. Considering that the Shackleton Ice Shelf, Pobeda Ice Island and Scott Glacier are all downstream of Mill Island, changes in these ice-scape elements are unlikely to have any strong influence on the Mill Island record, which is strongly influenced by changes to the east. Whereas the ice shelf configuration to the immediate west of Mill Island varies on decadal (or longer) timescales, the ice-scape to the immediate east (i.e., the region of ocean between Mill Island and Bowman Island) likely varies on much shorter timescales due to inter-annual variations in
SIC (see Section 5.3.2).

### 5.4 Sea salt source

Understanding the mechanism behind the observed high sea salt concentration is the key to further interpretation of the Mill Island record. Since wind speed does not strongly relate to the sea salt record here (as shown in Section 5.3.1), sea spray from the open ocean is unlikely to be the main sea salt source. The presence of negative $nssSO_4^{2-}$ values in the Mill Island ice core record (Fig. 6) indicates the occurrence of sea salt fractionation (i.e., a depletion of sulphate relative to sodium (Wagenbach
et al., 1998a)). The Mill Island sulphate record is highly fractionated (Appendix A) which indicates that frost flowers are likely to be an important sea salt source at Mill Island. Combined with the low altitude at the site and proximity to the sea salt source, frost flower-enriched aerosols may explain the high sea salt concentration at Mill Island.

Another factor contributing to the observed high sea salt levels may be rime accretion associated with fog events. When
supercooled fog droplets deposit onto a surface, it forms rime. Rime deposits generally have greater concentrations of all trace elements than fresh snow samples (Ferrier et al., 1995). Figure 21 shows an example of accumulated rime. This photo was taken on $23^{rd}$ October, 2011, at Roosevelt Island (79°25' S, 162°00' W), within the Ross Ice Shelf. Roosevelt Island is the ice core drilling site of the Roosevelt Island Climate Evolution (RICE) project (Tuohy et al., 2015). Roosevelt Island has a similar geographical setting to Mill Island, i.e., the distance from coast is ~20 km and the altitude of the summit is ~560 m. The field
team found ~0.5 m of rime ice on the Automatic Weather Station (AWS) when they returned to the site after an interval of one year (Fig. 21). The team experienced frequent fog, and growth of rime ice associated with the fog. The team also collected and analysed surface snow precipitation samples from the site. They found complex chemical signals such as multiple peaks of most measured trace elements within a single annual layer (A. Tuohy, personal communication, 2015).



With this in mind, the hypothetical snow ramp scenario proposed earlier, which explains the extremely high observed sea salt concentration, is developed further here.

New sea ice forms between Mill and Bowman Islands. Frost flowers form on the new sea ice. Also, upward migration of sea salt-enriched brine through sea ice produces salty snow on the sea ice. Frost flowers and salty snow are aerosolized then
transported west in the prevailing easterly wind. The coastal easterly wind also creates a coastal polynya in the lee of Bowman Island, allowing formation of more new sea ice, and so a constant supply of frost flowers can be produced. The easterly wind also facilitates formation of stable land-fast ice immediately east of Mill Island. Precipitation and drifting snow create a snow ramp which bridges the vertical discontinuity between the land-fast ice and the ice cap at the edge of Mill Island. Transport of frost flower and sublimed salty snow aerosols to the summit of Mill Island is facilitated by the presence of the snow ramp.
Alternatively, fog events may lead to rime accretion at the Mill Island summit.

Given the lack of in-situ chemical and physical observations at the eastern base of Mill Island, it is difficult to prove this hypothesis. For further study, a high resolution snow pit study and AWS will be crucial to verify this hypothesis.

## 6   Conclusions

The Mill Island ice core was dated by counting annual layers of $\delta^{18}O$, with support of the $H_2O_2$ and Deuterium excess (D-ex)
records as required. The ice core contains 97 years of climate record (from 1913 to 2009 C.E.). The dating uncertainty is + 2.4, − 3.5 year. The trace ion chemistry record of the Mill Island ice core was investigated by comparison with other nearby ice cores and instrumental data. The mean concentration of all major ion species except nitrate is much higher than in other nearby ice core records, e.g., Law Dome Summit South. In particular, sea salt concentration ($Na^+$ and $Cl^-$) is remarkably high (254 and 290 $\mu Eq/L$, respectively).

The Mill Island ice core record is characterised by a unique chemistry record with a mixture of clear and ambiguous seasonality, and high and low trace ion concentration periods with regime changes in 1934 and 2000 C.E.. The stratigraphy shows crust layers throughout the ice core. The cause of the crust layers is likely wind-related, and there is no evidence that the crust layers are caused by melt events. Furthermore, these layers are not the cause of the ambiguous trace ion seasonality.

Sea salt ions (particularly $Na^+$, $SO_4^{2-}$, and $Mg^{2+}$) were investigated in conjunction with records of environmental conditions
around Mill Island. It was found that the dominant wind direction is from the east, but wind speed was unlikely to influence the $Na^+$ and $SO_4^{2-}$ records at Mill Island. Instead, the $Na^+$ and $SO_4^{2-}$ records were found to correlate well with sea ice concentration between Mill and Bowman Islands. Based on current knowledge, no documented historical ice configuration changes have been noted that might affect the Mill Island ice core record. However the abrupt change in the sea salt record in 1934 may indicate a significant, unknown ice configuration change east of Mill Island.

A hypothetical mechanism for high sea salt concentration deposition was proposed, including sea ice concentration, snow ramp formation, and rime (associated with fog) deposition. Further studies, including installation of AWS at Mill Island and a high resolution snow pit study are required to prove this hypothesis.





## Appendix A: Non sea salt sulphate and fractionation

The presence of negative $nssSO_4^{2-}$ values in the Mill Island ice core record indicates that the calculation of $nssSO_4^{2-}$ is not accurate when using the $k$ value from typical seawater composition. A new $k$ value, $k'$, was calculated following Hall and Wolff (1998)

$nssSO_4^{2-}$ (calculated using the sea water ratio of $Na^+$ to $SO_4^{2-}$) versus $Na^+$ data are shown in Fig. 22. Negative $nssSO_4^{2-}$ concentrations extend down to $\sim -600$ *μEq/L*. The new, corrected $k$ value, $k'$, is obtained by subtracting the regression slope, $r$ from $k$, i.e., $k' = k - r$. As a result, $k' = 0.049$. The negative $nssSO_4^{2-}$ values from the 97 year ice core record were used for the regression line calculation, rather than only using winter data (Hall and Wolff, 1998). This methodology is due to the ambiguous seasonality in the Mill Island ice core record. The $nssSO_4^{2-}$ record during known volcanic eruption years (1991,

1984, and 1964) was excluded.

To verify this result, $k'$ was obtained independently by minimising the correlation coefficient between $nssSO_4^{2-}$ (calculated using a $k'$ range from 0.01 to 0.12 in steps of 0.001) and the associated $Na^+$ value (Wagenbach et al., 1998a). The correlation coefficient $r$ is a function of $k'$. The $k'$ value for a correlation coefficient of $r \cong 0$ is at $k' = 0.049$, confirming the simple linear regression method. The zero correlation indicates that the sea salt influence on $SO_4^{2-}$ is removed. Comparing with the LD $k'$

value of 0.087 (Palmer et al., 2002; Plummer et al., 2012), the Mill Island ice core is more highly fractionated.

*Author contributions.*  MI did the trace ion chemical measurement, led the analysis and wrote the manuscript. MC, AM and TVO provided expertise on ice core data interpretation and performed proof reading of the manuscript. ADF provided expertise on sea ice, atmospheric interpretation, and performed extensive proof reading of the manuscript. HP did proof reading of the manuscript. IG led the field work.

*Acknowledgements.*  The Australian Antarctic Division provided funding and logistical support (AAS1236). This work was supported by

the Japan Society for the Promotion of Science Grant-in-Aid for Scientific Research (KAKENHI) number 25·03748, and by the Australian Government's Cooperative Research Centre program through the Antarctic Climate & Ecosystems Cooperative Research Centre. The authors would like to thank to Ms Meredith Nation and Dr. Sam Poynter for their assistance in laboratory and Ms Indi Hodgson-Johnston for creating Fig. 16.



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



| Ice core | Lat | Lon | Depth (m) | Drill date |
|---|---|---|---|---|
| MI0910 | 65$^o$ 33' 10" S | 100$^o$ 47' 06" E | 120 | 2010-01-18 |
| MIp0910 | 65$^o$ 33' 10" S | 100$^o$ 47' 06" E | 10.57 | 2010-01-15 |
| MIp0809 | 65$^o$ 33' 25" S | 100$^o$ 33' 26" E | 16.69 | 2009-01-22 |

**Table 1.** Mill Island ice core information

| Wind speed | $Na^+$ | | | $SO_4^{2-}$ | | |
|---|---|---|---|---|---|---|
| (m/s) | annual | summer | winter | annual | summer | winter |
| < 5 | 0.22 | 0.11 | − 0.01 | 0.15 | 0.33 | − 0.09 |
| 5 − 15 | **− 0.51** | − 0.14 | **− 0.48** | **− 0.42** | − 0.32 | **− 0.36** |
| > 15 | 0.18 | 0.01 | **0.41** | 0.18 | − 0.09 | **0.42** |

**Table 2.** Correlations between CFSR wind speed at Mill Island and concentration of $Na^+$ and $SO_4^{2-}$ during winter and summer and for the annual average. Bold values indicate significant correlations (i.e., $p < 0.05$.)

| | $Na^+$ | $SO_4^{2-}$ |
|---|---|---|
| SIC-W | − 0.28 (0.129) | **− 0.49** (0.005) |
| SIC-C | **− 0.47** (0.008) | **− 0.52** (0.002) |
| SIC-E | **− 0.53** (0.002) | **− 0.55** (0.001) |
| SIC-S | **− 0.53** (0.002) | **− 0.58** (0.001) |
| SIC-SE | **− 0.57** (0.001) | **− 0.57** (0.001) |

**Table 3.** Correlation coefficients between annual mean sea ice concentration at each pixel and the annual mean concentration of $Na^+$ and $SO_4^{2-}$. Bold numbers indicate $p < 0.01$. The p-value is shown in brackets.



|  | $Na^+$ | $SO_4^{2-}$ |
|---|---|---|
| SIC-m$_{AB}$ | **− 0.56** (0.001) | **− 0.58** (0.001) |
| SIC-m$_A$ | **− 0.76** (0.017) | **− 0.80** (0.009) |
| SIC-m$_B$ | **− 0.52** (0.012) | **− 0.47** (0.029) |

**Table 4.** Correlation coefficients between annual mean sea ice concentration at SIC-m and the annual mean concentrations of $Na^+$ and $SO_4^{2-}$ during the periods of 1979 – 2009 (SIC-m$_{AB}$), regime A (2001 – 2009, SIC-m$_A$), and regime B (1979 – 2000, SIC-m$_B$). The p-value is shown in bracket. Bold numbers indicate $p < 0.05$.

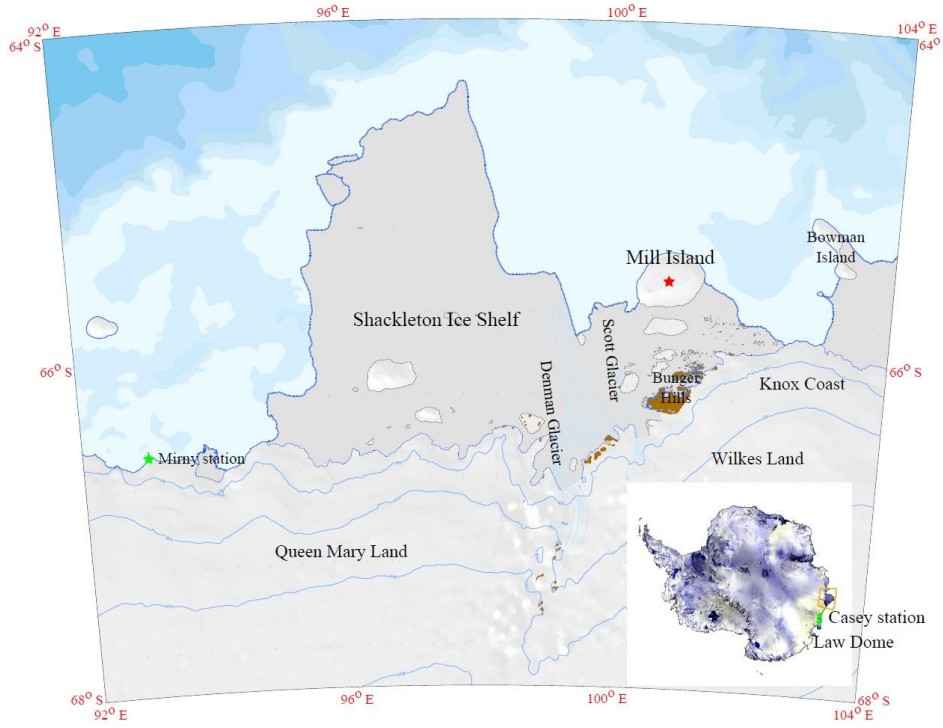

**Figure 1.** Location of Mill Island, East Antarctica. Mill Island is located adjacent to the Shackleton Ice Shelf, north of Bunger Hills. The red star shows the location of the 120m Mill Island ice core (MI0910). This map is modified from map number 13976 produced by the Australian Antarctic Data Centre, courtesy of the Australian Antarctic Division, © Commonwealth of Australia, 2012.

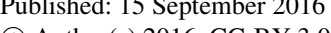


**Figure 2.** Comparison of (a) $\delta^{18}O$, (b) D-ex, (c) $Na^+$, (d) MSA, and (e) $SO_4^{2-}$ records from MI0910 (black solid line), MIp0910 (green dashed line) and MIp0809 (red dotted line).





**Figure 3.** Ninety-seven year record of H$_2$O$_2$ (a), $\delta^{18}$O (b), $\delta$D (c), and D-ex (d). All data were re-sampled to a 0.1 year grid and smoothed with a Gaussian filter of width $\sigma$=1 point.





**Figure 4.** Average seasonal cycles of a) $H_2O_2$, b) $\delta^{18}O$, c) $\delta D$, (d) D-ex for the entire MI0910 record. The error bars show the standard error of the mean.





**Figure 5.** Trace ion chemistry data: a) Na$^+$, b) Cl$^-$, c) MSA, and d) SO$_4^{2-}$. All data were re-sampled to a 0.1 year grid and smoothed with a Gaussian filter of $\sigma$ = 1 point. The Na$^+$ and Cl$^-$ records can be partitioned into three regimes (Regime A, B, and C).





**Figure 6.** Trace ion chemistry data: a) nssSO$_4^{2-}$, b) Mg, c) Ca, and d) NO$_3$. All data were re-sampled to a 0.1 year grid and smoothed with a Gaussian filter of $\sigma$ = 1 point.





**Figure 7.** Average seasonal cycles of a) $Na^+$, b) $Cl^-$, c) MSA, d) $SO_4^{2-}$, e) $nssSO_4^{2-}$, f) $Mg^+$, g) $Ca^+$ and h) $NO_3^-$. The error bars show the standard error of the mean.



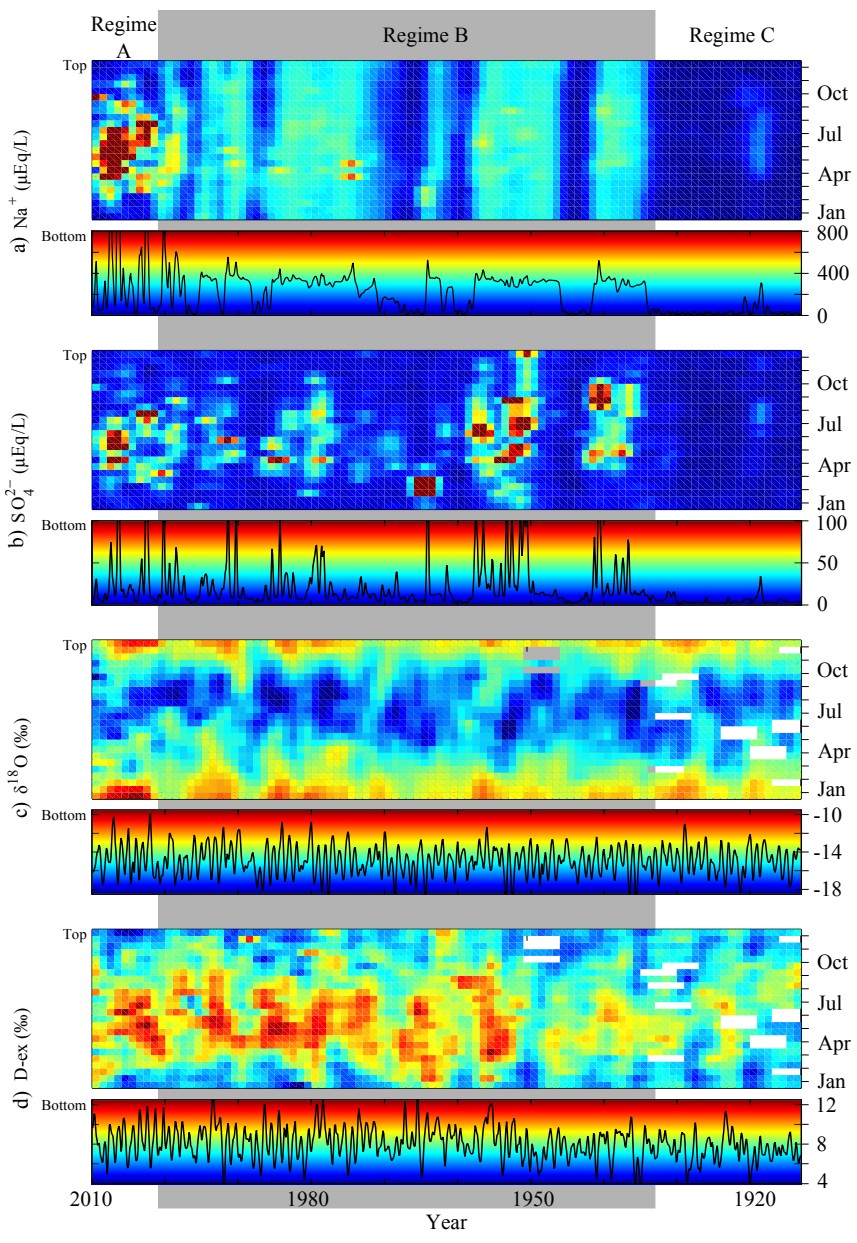

**Figure 8.** Time series of a) Na$^+$, b) SO$_4^{2-}$ concentrations, and c) $\delta^{18}$O, and d) D-ex ratios over the period from 1913 to 2009. Each top panel: Data were interpolated to 24 points per year, then smoothed with a Gaussian filter of width $\sigma = 1$ point. The x-axis is year, the y axis is month, and color scale is shown in each bottom panel. Each bottom panel: Time series for each species from Figs. 3 and 5. The background color indicates the color used in the top panel. Y axis is the concentration/ratio. Regime B (2000 – 1934) is shown using a grey panel to delineate the regime changes.



**Figure 9.** Average seasonal cycles of a) Na$^+$, b) SO$_4^{2-}$, c) $\delta^{18}$O, and d) D-ex for each regime. Regime A: 2001 – 2009 (blue), Regime B: 1934 – 2000 (green), Regime C: 1913 – 1933 (magenta). The x axis shows the month, y axis shows the concentration/ratio. Note that the Na$^+$ concentration is shown with a different scale for regime A (left y axis) and regimes B and C (right y axis).





**Figure 10.** Crust layers recorded in MI0910 ice core (blue vertical lines) with 97 years of $H_2O_2$, $Na^+$, and $SO_4^{2-}$ record. The thickness of the blue lines has been exaggerated, relative to the ice core thickness, in order to enhance visibility. Grey ellipses indicate regions discussed in the text. The firn/ice density is unrelated to the occurrence of crust layers.





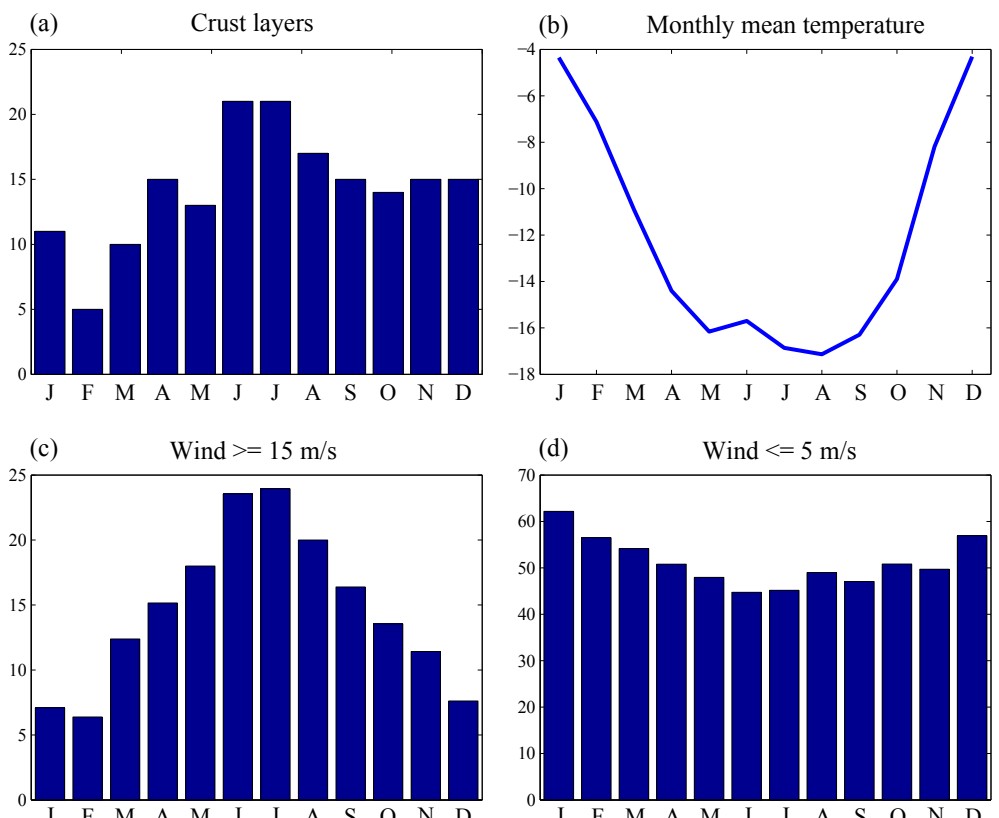

**Figure 11.** Figures of a) monthly total crust layers, b) Monthly mean temperature, c) monthly mean number of wind exceed 15 m/s data from six hourly data, d) same as c, but wind speed is less than 5 m/s.





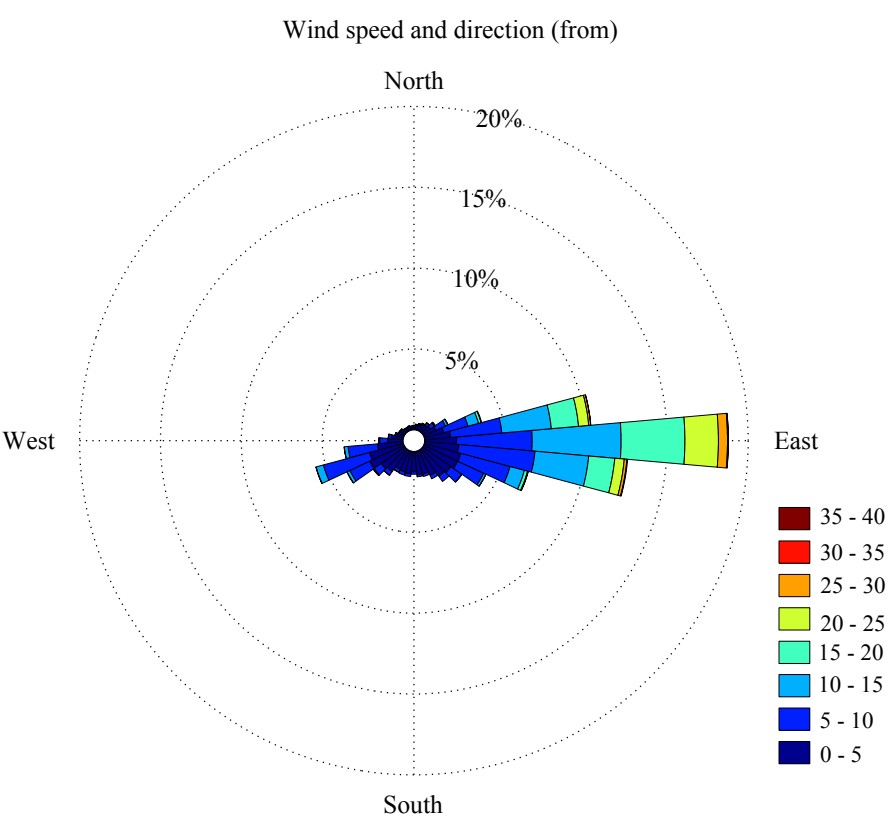

**Figure 12.** Wind rose climatology near Mill Island (CFSR grid point: 65.4119° S, 100.9375° E) from 1979 to 2009. The wind data were derived from the NCEP CFSR reanalysis model (Environmental Modeling Center, 2010).





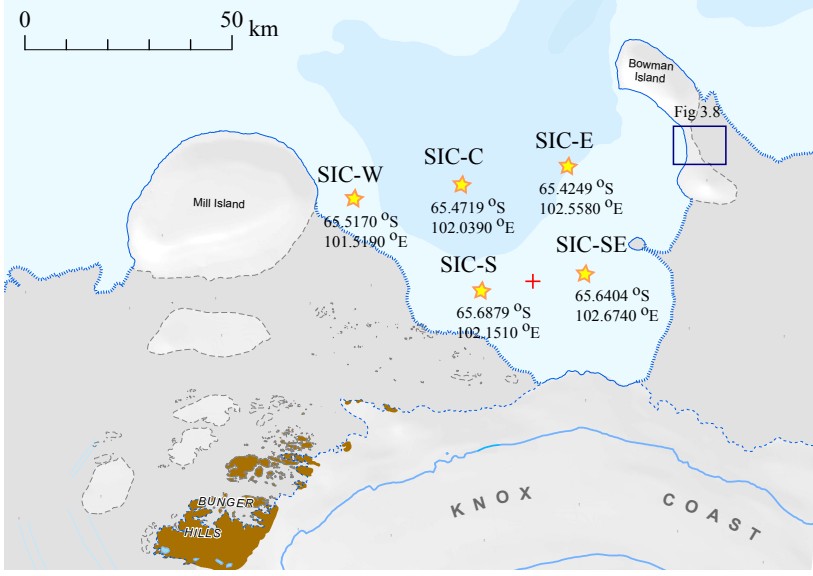

**Figure 13.** The coordinates and names of the five sea ice concentration data pixels. The red plus symbol indicates the centroid position of the derived time series SIC-m, formed by averaging SIC-S and SIC-SE. The dark blue rectangle indicates the location of the photograph shown in Fig. 17. Map courtesy of the Australian Antarctic Division, © Commonwealth of Australia, 2012.

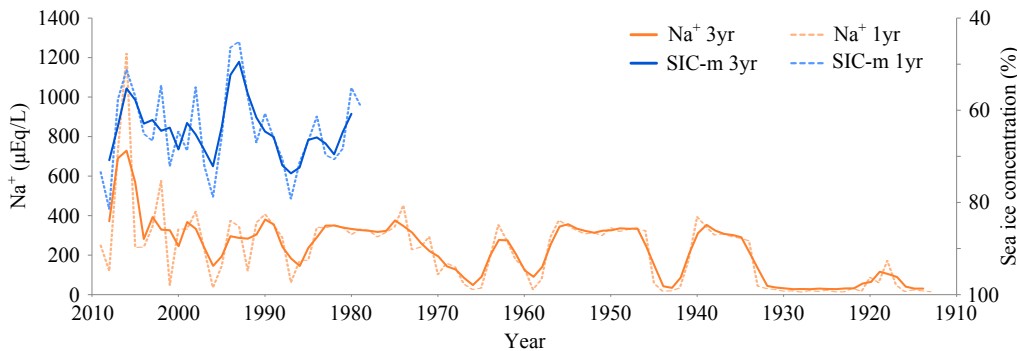

**Figure 14.** Time series of mean SIC-m (blue, right y axis) and, $Na^+$ (orange, left y axis) over the period from 1913 to 2009. Note that the right y axis is reversed to highlight the high degree of anti-correlation.





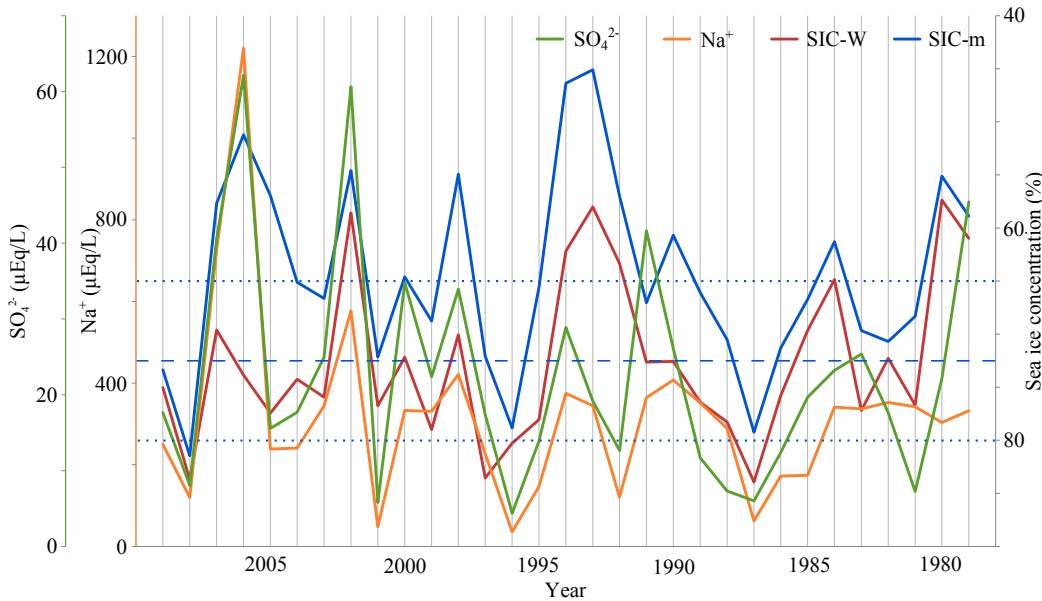

**Figure 15.** Time series of annual mean SIC-m (blue, right y axis), SIC-W (red, right y axis), Na$^+$ (orange, left y axis), and SO$_4^{2-}$ (green, left y axis) over the period from 1979 to 2009. The horizontal dashed blue line indicates the mean sea ice concentration in SIC-W, dotted blue lines indicate the $1\sigma$ standard deviation of the sea ice concentration in SIC-W. Sea ice concentration data were derived from NSIDC (see the text for details). Note that the right y axis is reversed to highlight the high degree of anti-correlation.







**Figure 16.** Schematic diagram of a hypothetical sea-salt transport mechanism at Mill Island, including formation of a snow ramp. (a) No land-fast sea ice case: Large sea salt particles cannot reach the Mill Island summit. (b) Land-fast sea ice case: Large sea salt particles can now reach the Mill Island summit.





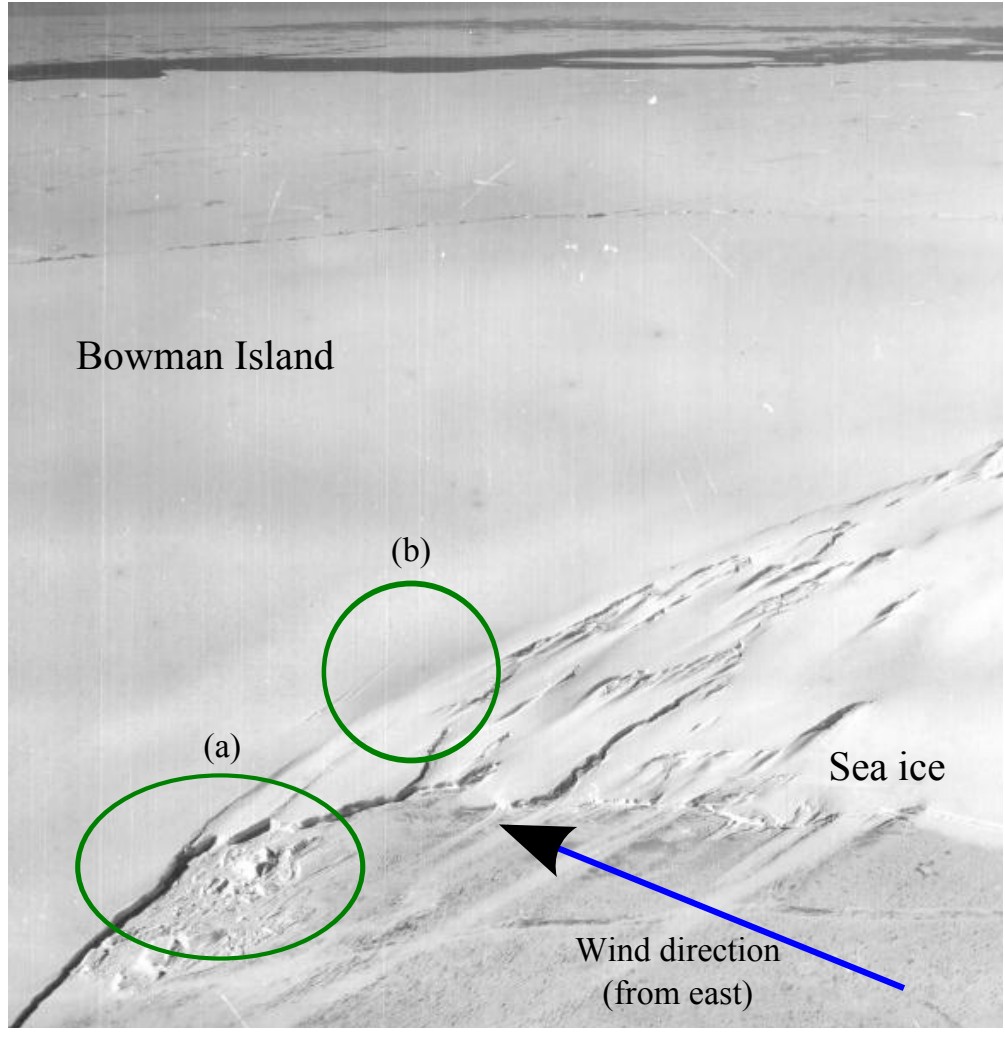

**Figure 17.** An aerial photo over Bowman Island on $11^{th}$ February, 1947. Ellipse a shows an example of the vertical discontinuity from sea level to the ice cap. Circle b demonstrates a clear snow ramp. Some scale is provided by cross-referencing with the rectangle in Fig. 13. Photo courtesy of the Australian Antarctic Division, © Commonwealth of Australia, 2015.







**Figure 18.** Annual variations in SIC-m, SIC-W, Na$^+$, and SO$_4^{2-}$ over the period from 1979 to 2009. The x axis is year, y axis is month, and the color shows sea ice/trace ion concentration. Each pixel shows the monthly mean concentration of associated species. Chemical data were interpolated to 12 data points per year. No filtering was used.





**Figure 19.** MODIS images of the Shackleton Ice Shelf and Mill Island area. The green dashed line divides the land-fast ice and the ice shelf. a) $14^{th}$ Feb 2002, b) $29^{th}$ Dec 2002





**Figure 20.** MODIS images of the Shackleton Ice Shelf and Mill Island area. a) $6^{th}$ Mar 2003, b) $15^{th}$ Sep 2004





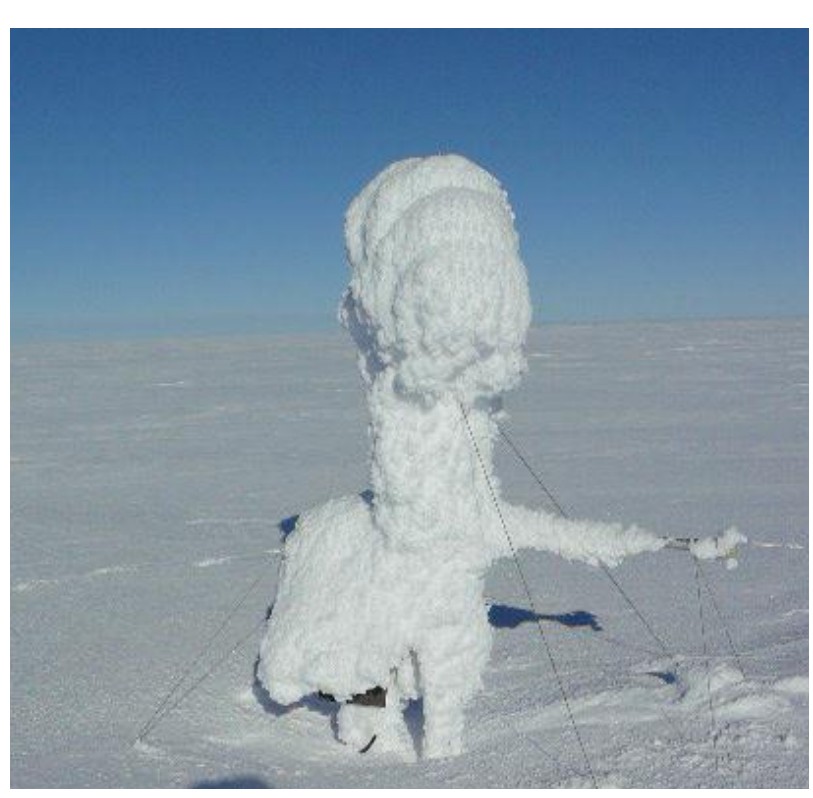

**Figure 21.** A photo of an Automatic Weather Station covered by thick rime ice at Roosevelt Island (79°25' S, 162°00' W), 23$^{rd}$ October, 2011. Photo provided by N. Bertler.





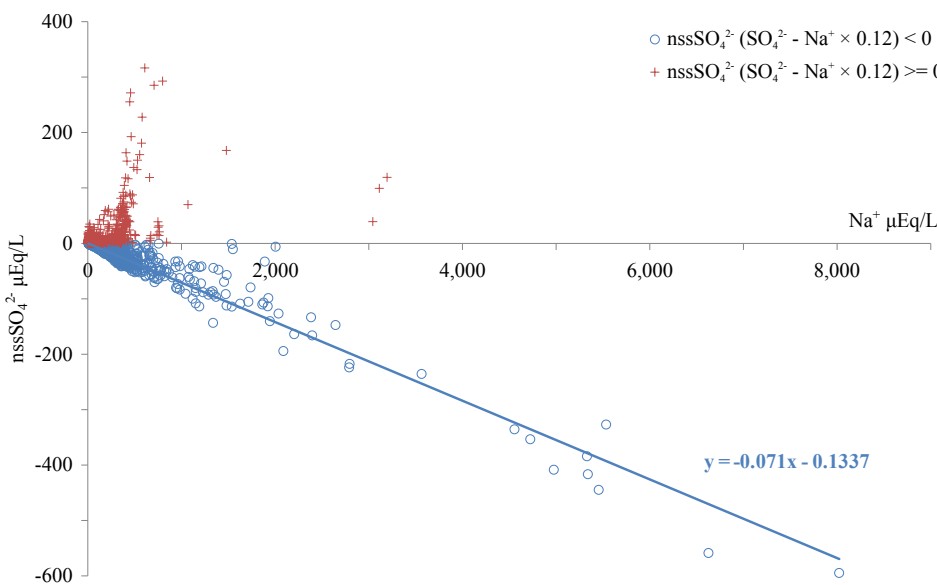

**Figure 22.** Scatter plot of nssSO$_4^{2-}$ (calculated using the seawater ratio of sulphate to sodium, 0.12) versus Na$^+$. The linear regression line was computed with negative nssSO$_4^{2-}$ values only.