# Peer review of "A glaciochemical study of the 120 m ice core from Mill Island, East Antarctica"

_Climate of the Past, 2016_

## Referee Comment (RC1) · Anonymous Referee #1 · 13 Oct 2016

This paper is an interesting and important contribution for understanding the environmental record of an ice core from a coastal site in East Antarctic. On the other hand, due to its low altitude and the number of thin crust/ice layers, I would suggest adding the following information in the text (to increase the confidence on the representativeness of the record):

(1) Ice core site mean temperature

It would be useful to know if the authors have measured the 10-15 deep temperatures in the snow pack, or at least have estimated the mean temperature at the ice core site.

The ice core site does not show evidences for strong post-depositional processes (i.e., partial melting, percolation and refreezing), but in the ionic ones ($Na+$ and $S04-2$) the seasonal variations seem to have been damped in some sections of the core (e.g.,

around 1980 and 1955 A.D. - Fig. 5 and 10). Further, some sections also show re-duction in the excess-deuterium (excess-d), this may be a further evidence of post-depositional melting and fractionation.

An examination of the Na+/Cl- relationship would give further evidence about the preservation of the original snow record.

By the way, what is the mean temperature at the near Mirny Station during January and February? Does it reach temperatures above 0°C, even only in some days?

Does any part of the core show signal of ionic preferential melting?

(2) Snow/ice density profile - ice stratigraphy

Considering the points above, and that the highest ionic amplitudes are found in the upper layers (after 1995), it would be useful to have a density profile in this paper.

It is quite common in cores of sites that suffer sporadic surface melting to have the original seasonal variations only in the upper layers (as it happens in the Mill Island core). Further down, melting, followed by percolation and refreezing, damp the signal.

The authors should at least report the thickness range of the observed crust/ice layers.

(3) Ice core dating

The authors tell that dating was confirmed by well-known volcanic eruptions (1991, 1984, and 1963, etc.). Please identify these eruptions in the SO4-2 profile (In Figure 2 and 6a).

(4) Figures

Figure 12 is redundant, please consider removing it.

---

## Referee Comment (RC2) · E. Isaksson (Referee) · 21 Oct 2016

The paper present and discuss the data from a 120 m deep ice core drilled from the coastal site Mill Island on the- Shackleton ice shelf, in East Antarctica. This is an ice coring site situated further north than most (if not all) other coring sites, except the sites on the Antarctic Peninsula. The drill site was chosen because of its coastal position with high accumulation rates and thus a good position to date and compare annually resolved data with both instrumental- and model output data. There is generally a lack of high resolution ice core data from Antarctica and therefore this drill site is of very high interest. The most intriguing results from this cores is evidence of migration of sea salt, something that has not been described in the literature before as I am aware of. Thus, the scope of the paper is highly interesting and should be published. However, the results are not easily accessible for the reader for reasons listed below. Before

the paper can be accepted there are some major reorganization and clarifications that are necessary. The paper is also too long and has far too many figures that are not necessary. Below I have tried to provide some guidance of how to re-organize the text and what to cut out so that the interesting results gets the visibility they deserve.

1. I suggest to combine "Results" and "Discussion" to get a better structure and save space (less repetition). Furthermore, try to eliminate the many short sub-chapters (there are now 3 levels) and integrate the text together with the longer sections. That makes the text more fluent for the reader.

2. Very little general glaciology and meteorology are provided-and the available information is not collected in one place so it makes it difficult for the reader. It would be good to collect the information that exists in a separate chapter - "area description"-in the beginning of the paper. For instance, the wind direction and wind speed information (chapter 5.3.1.) should be appearing in the general introduction because this is such fundamental information for all interpretation. Also, the text about the "Local ice shelf variability" (chapter 5.3.3) belongs in basic general description of the area rather than discussion.

3. Where is the firn-ice transition? Please provide information about the ice depth and calculation of the vertical strain rates used for correction of the annual layers

4. The accumulation record is not included or shown anywhere. It is a very crucial determining factor for the rest of the data interpretation that I absolutely need to be presented together with the ion data. At such a coastal site all the deposition can be assumed to be wet deposition and thus determined by the accumulation.

5. A follow-up questions to the previous comment: Could the variation in sea salt seasonality etc be related to accumulation and density variations?

6. The ice core site is rather specific from many points of view so a discussion in how representative the Mill Island record is for a wider area would be important to include.

Local systems seem to be more important for at least the accumulation than large-scale atmospheric processes. That raises the question to if this is also true for the other records from this core- something that is natural to ask because wet deposition is dominating.

7. Tables: I do not see the need for Table 2, Table 3 and Table 4 but I miss a table including all the glaciochemical data!

8. Figures: Several of the figures are not necessary. Figures 10, 12, 13, 16, 17, 19, 20, 21, and 22 should be removed since they do not provide and crucial information that the reader cannot read/understand from the text. Furthermore, I suggest to combine Fig 4 and Fig 7 and then Fig 5 and Fig 6.

---

## Author Comment (AC1) · 5 Dec 2016

Dear Anonymous referee #1

On behalf of my co-authors, I'd like to express my thanks for constructive comment on our article "A glaciochemical study of 120m ice core from Mill Island, East Antarctica". We were happy to receive your suggestions to improve our manuscript.

Please find below a response to each comment. We feel that the manuscript has been enhanced by incorporating these comments.

Kind Regards,

Mana Inoue

Responses to referee's comments:

[Figure]

1) Ice core site mean temperature

a. It would be useful to know the 10 to 15 meter depth snow pack temperature.

Text has been added on page 2, line 13. "The borehole temperature observed during the 2010/2011 summer is -13.86 Celsius, measured at a depth of 19.07 m from the 2011 C.E. surface (Roberts et.al., 2013)."

b. The ice core site does not show evidences for strong post-depositional processes, but in the ionic seasonal variations seem to have been damped in some sections of the core. Further, some sections also show reduction in the deuterium excess (D-ex). This may be a further evidence of post-depositional melting and fractionation.

We disagree with this comment. Crust layers in Figure 10 do not much with the D-ex reduction. The reduction in D-ex is due to sample resolution decrease with depth.

c. An examination of the Na+/Cl- relationship would give further evidence about the preservation of the original snow record.

Thank you. This is a good idea. However analysis of the Na+/Cl- relationship will be examined in a follow up paper which will be dedicated to post depositional migration of sea salt. As such, it is out of scope for this paper.

d. What is the mean temperature at the near Mirny Station during January and February? Does it reach temperatures above 0 Celsius even only in some days?

The average temperature at Mirny station during January and February is -1.84 and -5.25 Celsius respectively. It is possible to reach temperatures above 0 Celsius on some days. However, considering the ~500 m summit altitude Mill Island ice core site is much less likely to experience melt. Moreover, there is no evidence of significant melt in Mill Island ice core (see comment above).

e. Does any part of the core show signal of ionic preferential melting?

As in section 5.2.2., there are no evidence of melt in entire Mill Island ice core.

2) Snow/ice density profile – ice stratigraphy

a. Considering the points above, and that the highest ionic amplitudes are found in the upper layers, it would be useful to have a density profile in this paper.

Thank you for the good suggestion. A density profile has been added to the manuscript.

b. It is quite common in cores of sites that suffer sporadic surface melting to have the original seasonal variations only in the upper layers. Further down, melting, followed by percolation and refreezing, damp the signal.

This is true, but there is no evidence of melt from the density profile. The Mill Island density profile also matches the Law Dome density profile.

c. The authors should at least report the thickness range of the observed crust/ice layers.

Thank you. Crust layers observed in the Mill Island ice core ranges ∼1 to 5 mm thickness. Text has been added to the manuscript. "The stratigraphy of the MI0910 ice core shows ∼1 to 5 mm thickness of higher density layers distributed occasionally throughout the entire ice core."

3) Ice core dating

a. The authors tell that dating was confirmed by well-known volcanic eruptions. Please identify these eruptions in the sulphate profile.

Figure 6 (a) has been replaced to show non-sea salt sulphate calculated with modified k' as in Appendix A. Although nssSO42- from well-known volcanic eruptions do not significantly stand out in the Figure 6 (a), there are peaks during these volcanic periods. These results match with Law Dome nssSO42- results. The text has been changed in this section to reflect this. "Although nssSO42- peaks do not stand out for the major volcanic eruption, the timing of some nssSO42- peaks in MI0910 records matches with the eruption years."

4) Figures

a. Figure 12 is redundant. Please consider removing it.

Thank you. Figure 12 has been removed.

————————————————————

---

## Author Comment (AC2) · 5 Dec 2016

Dear Dr. Isaksson,

On behalf of my co-authors, I'd like to express my thanks for your constructive comment on our article "A glaciochemical study of 120m ice core from Mill Island, East Antarctica". We were happy to receive your suggestions to improve our manuscript.

Please find below a response to each comment. We feel that the manuscript has been enhanced by incorporating these comments.

Kind Regards,

Mana Inoue

Responses to referee's comments:

[Figure]

1) Suggestion to combine "Results" and "Discussion" to get a better structure. Other suggestion to eliminate the many short sub-chapters and integrate the text together with the longer sections.

Thank you for the suggestion. Short sub-chapters (2.3.1, Wind direction and wind speed; 2.3.2, Sea ice concentration; 5.2.1, Possibility of analytical error in measurement or methodology; 5.2.2, Possibility of snow/firn melt; 5.2.3, Possibility of true environmental signals; 5.3.1, Wind direction and wind speed at Mill Island; 5.3.2, Relationship between sea ice concentration and sea salt; 5.3.3, Local ice shelf variability) have been eliminated. However we have decided to maintain the Results and Discussion chapters as before because we think this enhances the flow of the paper.

2) Suggestion to collect the general glaciology and meteorology information in a separate chapter. For example, the wind direction and wind speed information (chapter 5.3.1) and the local ice shelf variability (chapter 5.3.3) should be in the general introduction.

Thank you for this suggestion. All general glaciology and meteorology information including wind direction and wind speed are now given in the introduction as a table. However, we've chosen to keep sub-chapters, 5.3.1 and 5.3.3 in the "discussion" as they are a part of the general discussion.

3) Provide information about the ice depth and calculation of the vertical strain rates used for correction of the annual layers.

This information is now included in the aforementioned general information table in the introduction.

4) Provide accumulation record.

As above: The annual accumulation rate is now included in the general information table. The full details of the accumulation record are out of scope for this paper.

5) Could the variation in sea salt seasonality etc be related to accumulation and density

variations?

The variation in sea salt seasonality does not appear to be related to accumulation and density variations. To demonstrate this, a figure of density profile (Fig. 9) has been included in the manuscript. The density profile matches well with Law Dome density profile.

6) How does Mill Island record represent local or wider area system?

Mill Island record is strongly inflenced by local conditions rather than wider area system. A statement has been included in the manuscript.

7) Tables 2, 3, and 4 are unnecessary. Adding a table including all the glaciochmical data would be more useful.

Thank you for the suggestion. The tables 2, 3 and 4 have been removed from the main text, and are now provided as supplementary information.

8) Figures 10, 12, 13, 16, 17, 19, 20, 21, and 22 should be removed, figures 4 and 7, 5 and 6 better be combined.

Thank you for the suggestion. Figures 12, 13, 17, 19, 20, and 21 have been removed and are now given as supplementary information. However, Figures 10, 16, and 22 have been retained in the manuscript as they are important for the discussion. Figure 4 has been removed and Figures 5 and 6 have been combined.

---

## Author Response (AR2)

Dear Dr Goto-Azuma,

On behalf of my co-authors, I would like to thank you for accepting our manuscript, "A glaciochemical study of 120 m ice core from Mill Island, East Antarctica", pending technical corrections.

Please find below a response to each technical comment.

Kind Regards,

Mana Inoue
* * *
**Responses to Editor's comments:**

1) **A snow accumulation record would add to the manuscript.**

   Annual snow accumulation record has been added in Figure 2. Related text has been added in the Method and Ice core dating sections. The additional text is "The Mill Island annual snow accumulation record was obtained from the thickness of annual layers after ice core dating was completed. The thickness of annual layers was corrected using the density profile and vertical strain rate, assuming that ice thinning is only caused by vertical strain. According to the method and density profile in Roberts et al. (2013), the vertical strain rate was estimated by least squares fitting from the density-corrected thickness of each annual layer. The flow correction is broadly consistent with the estimated ice thickness and a uniform vertical strain rate." And "Figure 2 presents annual snow accumulation, $H_2O_2$, $\delta\,^{18}O$, $\delta D$ and the D-ex records. The MI910 annual snow accumulation rate averages 1.35 mIE/yr for the period from 1913 to 2009, with a minimum of 0.79 mIE/yr in 1969 and a maximum of 2.04 mIE/yr in 1934 (Fig. 2 a)".

2) **The units for the snow accumulation record should be the same.**

   Thank you. The text has been changed from 1,312 kg/m$^2$yr to 1.430 mIE/yr.

3) **The full map in the Fig.1 needs to be larger.**

   Figure 1 has been fixed.

4) **Ion data which are not discussed in the text (K, Mg and Ca) need to be removed or explained in the text.**

   K has been removed from text. Mg and Ca have been explained in text. "Similarly, MSA, $SO_4^{2-}$, $Mg^{2+}$, and $Ca^{2+}$ show clear seasonality only for the period of 2009 to 2001."

5) **P. 4, L. 25. Pro-gram's. Is this correct?**

Thank you. This is a typo. It has been corrected as "Program's".

6) **P. 5, L. 1 – L. 2 and L. 24 – L. 26. State exactly which volcanic makers were used. Also mark them in Fig. 4.**

The volcanic makers are Pinatubo (1991), El Chichon (1982), and Agung (1963). These have been added in text and Fig. 4. "Arrows A, B, and C in the $nssSO_4^{2-}$ record correspond to major volcanic eruptions (Pinatubo [1991], El Chichon [1982], and Agung [1963], respectively)."

7) **P. 5, L. 33. "e.g.. Na…. has a summer peak" needs references.**

Thank you. A reference (Legrand and Mayewski [1997]) has been added.

8) **P. 6, L. 6 – L. 8. State how the months were defined.**

Months were computed by linearly dividing each year into 12 portions. This sentence has been added in text.

9) **P. 6, L. 18 states that "the Na winter peak druing regime A is not present in regime B", whilst L. 32 states that "Na and SO4 both show seasonality with a winter peak in Regimes A and B." Modify these sentences to avoid contradiction.**

Thank you. The sentence "the Na winter peak during regime A is not present in regime B" has been modified "the Na winter peak during regime A is less pronounced in regime B."

10) **References missing DOIs.**

Thank you. All DOIs have been added in references.

11) **Books should be more precisely referenced. n Fig. 6. Please correct the figure numbers.**

Thank you, the chapters have been added in references.

12) **P. 2, L. 21 and P. 16, L. 1. Wagenback et al., 1998a. should be Wagen bach et al., 1998.**

Thank you, it has been fixed.

13) **Supplement 1. L. 6, Fig. 7 should be Fig. S7.**

Thank you, it has been fixed.

**14) Supplementary Figs and Tables. Should be numbered as Fig. S# and Table S#.**

Thank you. It has been fixed.